# Coccidioidomycosis Osteoarticular Dissemination

**DOI:** 10.3390/jof9101002

**Published:** 2023-10-11

**Authors:** Benedicte M. Moni, Barton L. Wise, Gabriela G. Loots, Dina R. Weilhammer

**Affiliations:** 1Biosciences and Biotechnology Division, Lawrence Livermore National Laboratory, Livermore, CA 94550, USA; 2Lawrence J. Ellison Musculoskeletal Research Center, Department of Orthopaedic Surgery, University of California Davis Health, 2700 Stockton Blvd., Sacramento, CA 95817, USA; blwise@ucdavis.edu (B.L.W.);

**Keywords:** *Coccidioides*, coccidioidomycosis dissemination, skeletal infection, fungal osteomyelitis, fungal synovitis, arthritis, knee joint

## Abstract

Valley fever or coccidioidomycosis is a pulmonary infection caused by species of *Coccidioides* fungi that are endemic to California and Arizona. Skeletal coccidioidomycosis accounts for about half of disseminated infections, with the vertebral spine being the preferred site of dissemination. Most cases of skeletal coccidioidomycosis progress to bone destruction or spread to adjacent structures such as joints, tendons, and other soft tissues, causing significant pain and restricting mobility. Manifestations of such cases are usually nonspecific, making diagnosis very challenging, especially in non-endemic areas. The lack of basic knowledge and research data on the mechanisms defining susceptibility to extrapulmonary infection, especially when it involves bones and joints, prompted us to survey available clinical and animal data to establish specific research questions that remain to be investigated. In this review, we explore published literature reviews, case reports, and case series on the dissemination of coccidioidomycosis to bones and/or joints. We highlight key differential features with other conditions and opportunities for mechanistic and basic research studies that can help develop novel diagnostic, prognostic, and treatment strategies.

## 1. Introduction

Recently estimated at 350,000 infections annually in the United States, Coccidioidomycosis, also known as Valley Fever, continues to expand its clinical footprint from established endemic areas such as Arizona and California to several new hotspots in northern Mexico and select regions in Central and South America potentiating new endemic areas [1]. This expansion is likely a result of both climate and population changes [2,3]. This understudied disease is caused by inhalation of aerosolized spores of two dimorphic fungi, *Coccidioides* (*C.*) *immitis* and *C. posadasii.* While most cases (60%) are asymptomatic, resolve spontaneously, and fail to be accounted for in the clinic, ~40% of infections present a pulmonary disease ranging from a self-limited flu-like illness to more severe pneumonia [4]. Uncommonly, but instigating a significant burden on the health care system, 0.5–2% of cases progress to disseminated disease, especially among immunosuppressed individuals (e.g., transplant recipients, human immunodeficiency virus (HIV) patients, and pregnant women) and non-Caucasians (especially African Americans and Filipinos) [5]. The most common organs or sites for extrapulmonary coccidioidomycosis include the central nervous system (CNS), skin, bones, and joints. Skeletal coccidioidomycosis generally requires long-term medical treatment, with the possibility of relapse if not properly managed [4].

Although widely described in clinical reports as individual case studies, disseminated skeletal coccidioidomycosis is still poorly understood and remains uncharacterized mechanistically in animal models of dissemination. Most importantly, the genetic and molecular mechanisms that define susceptibility to a coccidioidal infection that progresses to disseminated disease rather than the typical self-limited pneumonia have yet to be determined. In this review, we will discuss disseminated coccidioidomycosis with an emphasis on skeletal infection and describe the clinical and animal studies that define the current understanding of skeletal coccidioidomycosis.

## 2. Methods

We conducted a thorough review of all published work on disseminated coccidioidomycosis or skeletal coccidioidomycosis involving the bones and/or joints (any year included) using the search terms “coccidioidomycosis”, “*Coccidioides*”, “osteomyelitis”, “arthritis”, “bone”, “joint”, “synovitis”, “coccidioidomycosis case series”, or “coccidioidomycosis case reports”. The search was limited to full manuscripts in the English language and eliminated conference abstracts. After examination, citations that were not relevant to the subject of this article were excluded. Of the remaining, 29 were case series, and 27 were case reports. Additional searches were focused on skeletal coccidioidomycosis in animals and general updates on current knowledge of skeletal coccidioidomycosis.

## 3. Extrapulmonary Coccidioidomycosis

Disseminated coccidioidomycosis is defined as *Coccidioides* infections that spread to organs outside of the pulmonary system and pleura space after the rupturing of spherules, causing the release of endospores that are carried by the blood or lymph stream to distal sites [5]. There are two types of dissemination: lymphatic and hematogenous, with the latter being more common. In the absence of antifungal therapy, disseminated infection can develop within two months after symptoms onset, or on some rare occasions, years later [6,7], and may have a mortality rate as high as 25% if left untreated, according to a retrospective study by Bays and colleagues [7].

Most cases of disseminated coccidioidomycosis involve one nonpulmonary site, although multisite dissemination, where two or more distinct systems or sites were affected, has been described and associated with decreased survival compared to single sites of infection [3,6,8,9,10]. Table 1 presents the distribution of case counts for the most common sites of dissemination as described by Adam et al. [6] in their retrospective analysis of 207 patients with disseminated coccidioidomycosis from the University Medical Center (UMC) in Arizona. While skeletal dissemination sites (axial: skull, vertebral column, and thoracic cage; peripheral: limbs, other bones) were not the most frequent sites of *Coccidiodes* dissemination (16.9%, 15.5% respectively), these sites are one of the most frequent sites associated with multisite infections (20.0%, 18.8%). In patients with disseminated coccidioidomycosis, the clinical manifestations varied widely and resembled many different unrelated conditions, making the diagnosis very challenging. For example, instances of pulmonary disease were observed in hematogenous coccidioidomycosis in 86% (38/44) of patients who exhibited diffuse nodules or bilateral pneumonia in their lungs [6]. Disseminated *Coccidioides* mostly affected the skin, the skeleton (bones and joints), and the central nervous system (CNS) (Figure 1) but was also identified at other anatomic sites such as lymph nodes, spleen, subcutaneous tissues, liver, kidneys, adrenal glands, and myocardium.

## 4. Risk Factors for Extrapulmonary Coccidioidomycosis

Certain groups of individuals exposed to *Coccidioides* species display higher rates of dissemination and are therefore considered to be at higher risk for severe disease. These groups include certain ethnic groups, patients with compromised immune status (patients on immunosuppressive medications, HIV patients, third-trimester pregnant women, and organ transplant recipients), patients with diabetes, tobacco smokers [10,13,14,15], and people aged over 60 years [6].

Higher rates of severe disease, including higher rates of dissemination, are associated with certain ethnic groups, notably those of African, Asian (especially Filipino), and Hispanic descent [5,13,16,17,18]. Some of the increased susceptibility may be explained by environmental and socioeconomic factors; however, differences in rates of severe and disseminated disease emerge even when controlling for these factors [19], suggesting that underlying host genetic factors play a role in disease susceptibility. Unraveling the contributions of specific genetic factors that contribute to the control of *Coccidioides* infection and how they vary within ethnic populations is an active area of study and will help inform how to treat, control, and prevent disseminated disease.

Late-stage pregnancy is considered a high-risk factor for severe or disseminated infection, presumably due to the combined effect of reduced cell-mediated immunity and the influence of sex hormones during pregnancy [15]. High levels of estrogen and progesterone during pregnancy were reported to stimulate *Coccidioides* spp. and favored the progression towards disseminated disease, which in turn contributed to a higher rate of maternal mortality if left untreated [20,21,22]. The management of coccidioidomycosis in pregnant women is critical yet very challenging as most antifungals, except Amphotericin B, are teratogenic [23].

Immunosuppressed persons, especially those with impaired T-cell function, were also shown to be more vulnerable to developing severe or disseminated coccidioidomycosis. Among these patients, individuals with diagnosed diabetes, inflammatory arthritis, hematologic/lymphatic malignancies (e.g., chronic lymphocytic leukemia), HIV, and those who have undergone solid organ transplantation were at elevated risk [24,25,26,27]. A recent report by Hsu and colleagues [28] linked the severity of coccidioidomycosis to the impaired recognition of β-glucan in some immunocompetent patients. β-glucan is a fungal polysaccharide found in most invasive fungal pathogens and is involved in tumor necrosis factor (TNF)-α and H_2_O_2_ production. Defects in fungal β-glucan recognition and response lead to reduced control of the infection from the host and, consequently, the development of severe and disseminated coccidioidomycosis.

The risk of severe and disseminated coccidioidomycosis was also elevated with certain medical therapies that are known to alter antifungal immunity, such as corticosteroids and TNF-α inhibitors [29,30,31,32]. TNF-α plays a central role in activating cellular immunity, including the regulation of cellular infiltration, activation, cytokine secretion, and the induction of cell death [33]. As such, it is critical to the control of *Coccidioides* in vivo [34], and thus, its inhibition can lead to more severe disease, including both a higher incidence of symptomatic pulmonary infection and an increased incidence of dissemination [35]. Dysregulation of TNF-α activity has been implicated in several autoimmune disorders, notably rheumatoid arthritis and inflammatory bowel disease, and there has been a proliferation of biologics and small molecules on the market that block its activity in recent decades [36]. As both the geographic spread of *Coccidioides* spp. and the average age of the population increases, the increased incidence of severe and disseminated coccidioidomycosis due to immunosuppressive therapy is likely to also rise. Concurrent antifungal treatment to prevent reactivation of dormant infection and/or prevention of progression of severe/disseminated disease during primary infection during immunosuppressive drug treatment may be required [32].

While few reports have established diabetes as a risk factor for disseminated coccidioidomycosis, most clinical and scientific reports have argued that diabetes is mostly associated with complicated pulmonary coccidioidomycosis rather than with disseminated disease [6]. In one retrospective review of 39 patients with vertebral coccidioidomycosis, Szeyko and colleagues [37] supported the idea that diabetes is not to be considered a risk factor for disseminated coccidioidomycosis, yet several clinical reports have mentioned diabetes mellitus as part of the medical history of the patients with disseminated coccidioidomycosis [3,8,38,39].

Although disseminated coccidioidomycosis affects all ages, it is most common in adults. However, additional studies on the role of age as a risk factor still need to be conducted to conclusively determine functional links [15,16]. Higher incidence rates of coccidioidal infection were documented among men than women. Previous reports have linked this observed gender disparity to occupational hazards, suggesting that men engage in outdoor activities or agricultural jobs at higher frequencies than women. The disseminated disease was also reported to occur at higher rates in men than in women, suggesting that there may also be a genetic or hormonal basis for this gender disparity [15,40].

## 5. Osteoarticular Coccidioidomycosis

### 5.1. Clinical Reports

Skeletal coccidioidomycosis generally arises from hematogenous dissemination and can often progress to the destruction of bones or adjacent structures such as joints, tendons, and other soft tissues [4,9]. Skeletal sites of infection do not have a location bias, as any bone could be involved in dissemination, but the reported sites with the most severe disease manifestation have been in the axial skeleton, including the skull, sternum, ribs, and vertebrae [4,37,41,42], with the latter being slightly favored (especially the lumbar and thoracic areas), where *Coccidioides* spp. frequently disseminate [5,18,37,41]. Most clinical cases describe vertebral coccidioidomycosis as osteomyelitis (limited to the vertebral bodies) and discitis (intervertebral disc space involvement), with symptoms including vertebral body compression and height loss, worsening back and neck pain, lower extremities weakness, weight loss, night sweats [37,39,43,44,45,46,47], epidural enhancement and abscess [37,39,45,47], paraspinal abscess [47], complete vertebral body destruction with focal kyphosis, and retropharyngeal abscess [44].

One complicating factor in the diagnosis of vertebral coccidioidomycosis is that patients do not always present symptoms of pulmonary coccidioidomycosis, nor do they display a significant past medical history of elevated rates of respiratory infections [39,44,46,48]. Although some reports describe the presence of primary lung infection in patients with vertebral coccidioidomycosis [39], a retrospective review study by Szeyko and colleagues [37] showed that the pulmonary disease, even when present, did not correlate with the severity of vertebral infection. When the infection of the skull occurs from vertebral coccidioidomycosis [49], it can spread to the pachymeninges and the subarachnoid space and cause neurologic compromise characterized by symptoms such as motor deficits, mechanical instability, neural compression, sensory disturbances, leptomeningeal enhancement, and arachnoiditis. The clinical studies reporting neurologic symptoms in patients with vertebral coccidioidomycosis cases have also highlighted the necessity of surgical management to prevent further spread in addition to pharmacological therapy [43,44,45,50] due to the fatal nature of coccidioidal infection in the brain.

Some rare reported cases of coccidioidal osteomyelitis included infection of the first toe in a young patient with a medical history of diabetes insipidus and obesity [48], facial bone involvement such as the jaw osteomyelitis in an infant [41], infection of the patella (anterior bone of the knee) in immunocompetent males [51], and orbital bone osteomyelitis with associated periorbital abscess [52]. Coccidioidal osteomyelitis in those rare sites highlights the importance of considering *Coccidioides* spp. As potential pathogens during the diagnosis of patients with bone diseases, especially patients living in endemic regions or patients with a travel history to endemic regions, whether they are immunocompetent or immunocompromised.

Skeletal coccidiomycosis cases often appear with nonspecific features, mimicking other conditions such as malignant tumors, tuberculous osteomyelitis, as well as bone infections caused by other microorganisms [4,9]. Caraway and colleagues described two different cases of coccidioidal osteomyelitis that were misdiagnosed as primary bone tumors [53]. In the first case, the patient with right hip pain was diagnosed with an epithelioid hemangioendothelioma after a core biopsy showed histiocytoid cells. For the second case, the patient presented a mass in the manubrium that was initially described as lymphoma with metastases based on radiographic findings. In both cases, a subsequent fine needle aspiration (FNA) biopsy helped correctly diagnose coccidioidomycosis osteomyelitis.

Similarly, vertebral coccidioidomycosis may also be misdiagnosed, being mistaken for spinal metastasis [54,55], tuberculous spondylitis (Pott’s disease: tuberculosis with spine involvement), pyogenic spondylitis (spinal infection from other bacteria origin such as *Staphylococcus aureus*) [37] or other granulomatous diseases [55]. One report by Wesselius et al. [56] examined a case of vertebral coccidioidomycosis that appeared with similar clinical and radiological features as those of vertebral tuberculosis yet indicated differences in distinguishing the conditions. Although both conditions can present with similar roentgenologic changes, coccidioidomycosis produces greater bone destruction with a rapid progression and symptom onset than tuberculous spondylitis, which appears with an indolent symptom onset [57]. The summary of distinctive clinical features for spinal metastasis, vertebral coccidioidomycosis, pyogenic and tuberculous spondylitis are presented in Table 2 [37,57,58,59].

Some cases of bone infections can spread from adjacent soft tissues rather than through hematogenous dissemination [60]. Soft tissue infections are relatively rare in Valley fever patients but can be very severe and often behave as invasive neoplasms [61,62]. *Coccidioides* dissemination to the bones often occurs with simultaneous involvement of the associated joints [4,41,63], with joint infection typically occurring via spread from the bone and rarely as the result of the hematogenous spread [64]. In their retrospective clinical analysis, Adam and colleagues [6] found that all 14 patients with joint disease exhibited evidence of skeletal disease in the bone adjacent to the joint, suggesting that the majority of patients with joint infection most likely present adjacent osseous infection. Nevertheless, although rare, cases of coccidioidal arthritis with no bone involvement exist. An example of coccidioidal synovitis localized to the knee joint with no osseous destruction was presented in the clinical report by Coba and colleagues [8].

The most common manifestations of joint dissemination are arthritis and synovitis with effusion, with the lower extremities, such as the knees, being the most affected articulated joint, followed by the ankles and then the wrists [5,65]. Ahmad and colleagues [65] reported a well-managed case of knee joint effusion and synovitis, with no bone involvement, caused by *C. immitis* and *C. posadasii* in a 49-year-old man from Texas. This case was resolved after antifungal treatment together with synovectomy, arthrotomy, and drainage without further relapse after a six-month follow-up. Although very rare, coccidioidomycosis dissemination can also occur at prosthetic knee joints and cause severe infections, which are generally mistaken for being of bacterial origin [66,67].

In addition to CNS coccidioidomycosis, cases of coccidioidomycosis with the involvement of both joint and bone (osteoarthritis) are generally classified as more severe, and the contiguous bones often represent potential sites of relapse. Their management may require a variety of surgical procedures, including incision, drainage, surgical debridement, synovectomy, as well as bone resection, in addition to lifelong antifungal treatment [68]. One report on coccidioidal infection of the knee of a 62-year-old man has shown the chronicity of this fungal infection, evaluated during 24 years of follow-up with four different relapses, the longest one occurring after 11 years [69]. It was only on the fourth relapse that advanced diagnosis showed the presence of existing bone infection, suggesting a possible site of relapse. The case was finally resolved after debridement and synovectomy, followed by an arthrodesis with an external fixation and long-term azole therapy.

While most cases of joint coccidioidal septic arthritis appear in only one site of infection, such as the case described by Weisenberg [63], in which the patient, a 78-year-old man, had no other systemic symptoms apart from knee pain and swelling, in a minority of patients the dissemination of coccidioidomycosis can involve more joints. Nasrawi and colleagues [70] described a case of coccidioidal polyarticular septic arthritis involving the right wrist, left elbow, left ankle, and left knee joints, as well as the associated bones and soft tissues. The patient was initially diagnosed with chlamydia infection, which led to a misdiagnosis of reactive arthritis, a painful inflammation of joints occurring as part of the response to infection by *Chlamydia trachomatis*. The involvement of many joints resulted in this case being mistaken for desert rheumatism, a condition affecting joints bilaterally and causing a symptom called erythema nodosum. The absence of both the bilateral joint involvement and the symptom of erythema nodosum in that case, and the fact that the culture of fluids obtained from the knee, elbow, and wrist joints grew *C. immitis*, directly excluded the possibilities of reactive and rheumatoid arthritis. The management of the case was made possible by the application of appropriate antifungal and antibacterial therapy. This case report presents the challenge of managing cases that manifest with multiple pathologic options and suggests that coccidioidomycosis infection should always be considered as a potential cause in monoarticular or polyarticular septic arthritis, especially if patients are linked to Valley Fever endemic regions.

### 5.2. Diagnosis

Diagnostics for coccidioidomycosis have recently been thoroughly reviewed in [71]. Culture or isolation of *Coccidioides* spp. is considered the gold standard for the diagnosis and confirmation of coccidioidomycosis in suspected Valley Fever patients [14,65,72]. Culture of clinical specimens from any location can confirm the fungal infection; however, when skeletal dissemination is suspected, isolation of the *Coccidioides* spp. followed by a histopathologic examination performed on the involved bone, obtained by either percutaneous biopsy, CT-guided needle biopsy, or surgical debridement. In the case of joint involvement, the culture is done from joint fluid obtained by arthrocentesis [4]. Histopathologic testing can be assisted by molecular techniques such as in situ hybridization (ISH), which can bring more specificity to tissue analysis. Other techniques, such as polymerase chain reaction (PCR), have also been considered recently to assist the histopathologic examination [73] and confirm the presence of specific pathogen species. He and colleagues [74] addressed the question of whether decalcification of skeletal tissues from formalin-fixed paraffin-embedded (FFPE) samples would affect the sensitivity and specificity of quantitative PCR. According to their report on the detection of skeletal tuberculosis, the sensitivity and specificity of PCR were not impacted by decalcification and were even higher when compared to the acid-fast stain detection method. Moreover, Pandey et al. [75] reported that the detection of tuberculosis by the PCR method from different tissues, including bone tissues, showed a sensitivity of 73.07% and a specificity of 93.75%. PCR methods can, therefore, be added as part of routines in the detection of coccidioidomycosis to enable rapid detection and avoid misdiagnosis and delay in patient management.

Another possible method for earlier detection of coccidioidomycosis is the detection of β-D-glucans (BDG), a component of the cell wall of many fungal pathogens, which can be very useful, especially as an adjunct test for immunocompromised patients with coccidioidomycosis, who usually have no detectable specific antibody titers [76]. A significant disadvantage of this method is the reduced sensitivity in immunocompetent patients with acute coccidioidomycosis, which makes it less reliable for diagnosis [77,78].

Serological testing methods for confirmation of coccidioidal infection include enzyme immunoassays (EIA), often used to screen for coccidioidomycosis in high volume settings, immunodiffusion, which can detect both IgG and IgM antibodies despite its sensitivity varying with serum concentration, and complement fixation (CF) testing, which detects IgG antibodies to the chitinase antigen [41,79]. CF testing is the most commonly used serologic method due to its specificity and correlation with disease severity. CF titer in patients with disseminated infections is generally higher than 1:16, and its variation can inform the therapeutic response in patients or the disease relapse [14,41]. Extremely high CF titers that were defined as predictive of disease prognosis, dissemination, and complicated infections requiring surgical intervention in the preantifungal era [4,80], were reconsidered in patients treated with antifungals. According to the work published by McHardy and collaborators, with the availability of antifungal treatments, high CF titers were also found in patients with uncomplicated pulmonary infections and non-disseminated pulmonary infections, suggesting that decisions about patient management should not be based on CF alone but rather on the clinical presentation of patients and the risk factors they may present [79].

Although imaging can sometimes result in misdiagnosis as other conditions, such as cancer, bacterial infections, and fungal infections, it remains a very useful tool in the evaluation of the extent of skeletal dissemination and the necessity of surgery [4,65]. While radiographs can show bony lesions at the late stage of the disease progression, CT scans can visualize bone abnormalities with more specificity at any stage of the disease. MRI is generally used to orient the decision of surgical debridement by identifying the abnormal areas with more precision and clarity, the damage to soft tissues, and the abscess formation [4,38,72]. In the case of radiography and MRI failure, the technetium-99m (^99m^Tc) bone scan is the imaging method of choice due to its high sensitivity and ability to determine multicentric osteomyelitis [4,81].

Some reports of skeletal coccidioidomycosis have highlighted the potential diagnosis complications that can arise from the presence of additional superimposed bacterial and fungal infections in patients with pre-existing *Coccidioides* infection. In the case of coccidioidal polyarticular septic arthritis mentioned above as described by Nasrawi [70], the fluid from the patient’s wrist grew Methicillin-resistant *Staphylococcus aureus* (MRSA), in addition to *C. immitis*, making the patient’s condition significantly more difficult to manage. In a similar case, a 27-year-old patient from Ohio, who visited Arizona for six months, was initially diagnosed with cervical spine infection due to *C. immitis* and was found 13 months later after the surgery with methicillin-susceptible *Staphylococcus aureus* (MSSA) and *C. immitis* co-infection of the neck with further lesions at multiple vertebral bodies, the ribs, and the sacrum. A few months later, the infection by the two microorganisms extended to the sacrum, the ilium, and the acetabulum [44]. Another example of *Coccidioides* co-infection with other microorganisms is the case described by Fraser and colleagues [82], in which a 52-year-old man initially diagnosed with pulmonary tuberculosis, developed swelling and infection on his right calcaneus and left tibia from which wound-draining fluid grew *Mycobacterium tuberculosis*. Further examination of biopsies from the soft tissues around the infected bones showed *C. immitis* that he had probably contracted when he traveled to Arizona a few years before diagnosis. Nassif and colleagues [83] reported a case of disseminated skeletal coccidioidomycosis in a 67-year-old woman in the southwestern United States, previously diagnosed with COVID-19. After the resolution of COVID-19, the respiratory symptoms persisted, and imaging revealed bone infection in the chest wall, mimicking metastatic disease. Subsequent biopsies from the chest wall confirmed coccidioidomycosis, and the patient was put on oral fluconazole. This case highlights that potential co-infection with *Coccidioides* spp. and SARS-CoV-2 in COVID-19-positive patients, especially in endemic areas, should be considered as part of the diagnostic routine.

These clinical findings demonstrate that coccidioidal dissemination, in particular skeletal dissemination, can mimic many other diseases, making the diagnosis very challenging and unspecific [61,82,84]. Differentiating coccidioidomycosis infection from those other diseases is of utmost importance, given the therapeutic and prognostic implications. Accurate diagnosis of skeletal coccidioidomycosis requires a very high level of rigor and a multidisciplinary approach combining laboratory analysis (histopathology, immunology, and microbiology) and imaging (such as radiographs and MRI), especially in immunocompetent patients, and people from endemic areas [9,41,65,72].

### 5.3. Management of Osteoarticular Coccidioidomycosis

Effectively managing osseous and/or joint coccidioidomycosis involves several key factors: early detection, selecting the right treatment, making timely decisions regarding surgery, and ensuring thorough follow-up care. Current guidelines for the treatment of coccidioidomycosis recommend azole therapy for bone and/or joint dissemination [85]. Itraconazole is preferred over fluconazole for skeletal diseases [39,63,65,86]. However, for good bone penetration, posaconazole was found superior to itraconazole [87]. Amphotericin B is an alternative therapy recommended for patients who poorly respond to azoles and for severe osseous diseases, especially in critical bone locations such as the vertebral column [39,88]. Table 3 contains a summary of the indications, advantages, and side effects of the various antifungals used for the treatment of disseminated coccidioidomycosis in the bones and joints. Variability in the dose-exposure relationship, oral absorption, interactions with other drugs, toxicity, etc., resulting in the alteration of antifungal serum concentrations can be a particular challenge with azole therapy [89]; thus, therapeutic drug monitoring (TMD) has been strongly recommended to ensure patients maintain appropriate serum concentrations for better clinical outcomes [90]. Severe bone infections often necessitate aggressive surgical interventions, the delay of which can result in neurological compromise and deformity in patients with spinal instability and vertebral injuries [91]. The clinical improvement of severe bone infections can also be obtained with the use of surgically implanted antifungal-impregnated beads, which offer effective release of antifungals directly in the infected sites [92]. Fluconazole-impregnated beads are an effective strategy for the treatment of coccidioidal osteomyelitis [68]. A newer antifungal Olorofim has shown promise against several difficult-to-treat fungal infections, including disseminated coccidioidomycosis [93], and may play a larger role in the treatment of osteoarticular disease in the coming years, particularly against infections that display resistance against standard antifungal treatments.

The risk of recurrence and dissemination of coccidioidomycosis can be reduced with an early diagnosis, effective systemic antifungal therapy, regular and appropriate follow-up, adjusted according to the severity of the disease, the risk factors, and the treatment plan of each patient. Follow-up usually consists of regular clinical and imaging visits to assess the CF titers, the presence of new symptoms, probable adverse effects, the need for surgery, as well as TMD for azoles when needed [5,6].

### 5.4. Animal Studies

The necessity of prolonged antifungal treatment, the risk of relapse, and repetitive surgery remain a heavy economic burden for patients with skeletal coccidioidomycosis. It is essential to establish reliable animal models of infection to gain insights into the processes underlying the spread of *Coccidioides* spp. to the bones. Additionally, these endeavors will assist in the creation of novel approaches for preventing, diagnosing, and treating skeletal coccidioidomycosis.

Whereas reports have been published describing disseminated infection following naturally acquired infection of various domesticated mammalian species, such as dogs [94,95], cats [96], horses [97], llamas, and alpacas [98], there is a paucity of experimental small rodent models for studying disseminated infection. Most mouse studies of coccidioidomycosis have focused on localized infection (pulmonary disease) rather than the disseminated disease. To mimic human dissemination, the intravenous and intraperitoneal routes of infection have been used, although these generally require large numbers of arthroconidia to establish infection and are less lethal than pulmonary infection [99,100]. The few existing murine studies on the dissemination of coccidioidomycosis have focused on the spleen, lungs, liver, kidneys, and brain. These include studies of the immune response to systemic *C. immitis* infection of the lungs, spleen, and liver [101]; *C. immitis* tissue invasiveness in lungs, liver, spleen, and kidneys [102] as well as the drug efficacy against disseminated coccidioidomycosis in lungs, spleen, liver, and CNS [103,104,105]. Investigations using rodent animal models to study the dissemination of coccidioidomycosis, or other fungal infections, to bones and joints are lacking.

One of the challenges in studying disseminated infection in mice is the extreme virulence of *Coccidioides* infection in most inbred mouse strains. Intranasal infection of mice typically results in lethal pneumonia developing within 2–3 weeks; thus, there is insufficient time to develop the type of dissemination seen in human patients [106]. A recently described strain of *C. posadasii* displays growth delays in vitro and in vivo in mice, allowing for the infection to be assessed over a longer period. This strain has been proposed as a novel means for studying chronic infection [106,107]. Utilization of this *C. posadasii* strain, in combination with peripheral infection routes such as intraperitoneal or intraarticular injection, may provide new opportunities to study *Coccidioides* infection of the bones and/or joints in mice.

In contrast, several mouse models have been developed to study different aspects of bacterial osteomyelitis, such as prophylaxis, pathogenesis, and treatments, with *Staphylococcus aureus* being the main bacterial species used [108]. In their review, Guarch-Perez and colleagues [108] have categorized the existing mouse experimental studies on infectious osteomyelitis into five categories, namely, hematogenous osteomyelitis, post-traumatic osteomyelitis, bone-implant-related infection, peri-prosthetic joint infection, and fracture-related infection.

The mouse experimental studies on hematogenous osteomyelitis have mostly focused on *S. aureus* and used the intravenous route of infection to mimic the human infection dissemination through the bloodstream. Results from these studies have elucidated several mechanisms of *S. aureus* pathogenesis and related pathways, such as internalization mechanisms into osteoblasts and the involvement of EGFR/FAK and c-Src pathways [109]; mechanisms and pathways involved in triggering persistent and chronic osteomyelitis [110], as well as bone loss [111]. A similar application of animal models for the study of osteoarticular coccidioidomycosis dissemination will provide important insights into the complex pathophysiology of skeletal coccidioidomycosis and eventually lead to the development of more treatments and prevention strategies.

### 5.5. Conclusions

Manifestations of *Coccidioides* dissemination to bones and/or joints are often nonspecific. Although existing knowledge of *Coccidioides* infection and the patient’s medical history may be helpful, the initial diagnosis of osteoarticular dissemination can be difficult and is often confounded by other inflammatory conditions and infectious diseases. Successful management of osteoarticular coccidioidomycosis will, therefore, require a very high index of suspicion. Confirmation of the diagnosis is obtained either by histopathologic identification or by cultures, methods which delay patient management and promote dissemination to other organs, including difficult-to-treat bone and joint infections. Early detection will require the use of methods that deliver the results in a timely, efficient manner, such as PCR and genomic analysis, which, although available, are not yet widely used. In addition, imaging tests may show abnormalities and provide information on the existence of a bone infection but cannot distinguish osteoarticular coccidioidomycosis from other bone infections. To avoid confusion, they should be used as secondary tests to confirm coccidioidal infection, but not as primary tests.

The lack of a murine model recapitulating the dissemination of *Coccidioides* spp. to bones and joints has hampered the understanding of the mechanisms involved in dissemination and penetration to these sites. Establishing a murine model that mimics osteoarticular coccidioidomycosis using mouse strains of differing susceptibility can help identify not only the kinetic patterns of infection and bone/joint destruction and the fungal load required to induce lesions but also the fungal virulence or host resistance factors that are involved in the pathogenic process. Moreover, although they cannot reproduce the complexity of the in vivo environment, in vitro assays of the interactions of lung cells and *Coccidioides* species can provide insight into fungal pathogenesis, possible pathways involved in the dissemination to other organs, and factors defining the tropism for targeted organs of the extrapulmonary infection.

Finally, a high level of discernment would help in identifying coccidioidomycosis among other respiratory diseases of similar presentation, especially in patients with a history of travel to endemic areas, which may also lead to a swifter diagnosis of disseminated disease, including skeletal coccidioidomycosis.

## Figures and Tables

**Figure 1 jof-09-01002-f001:**
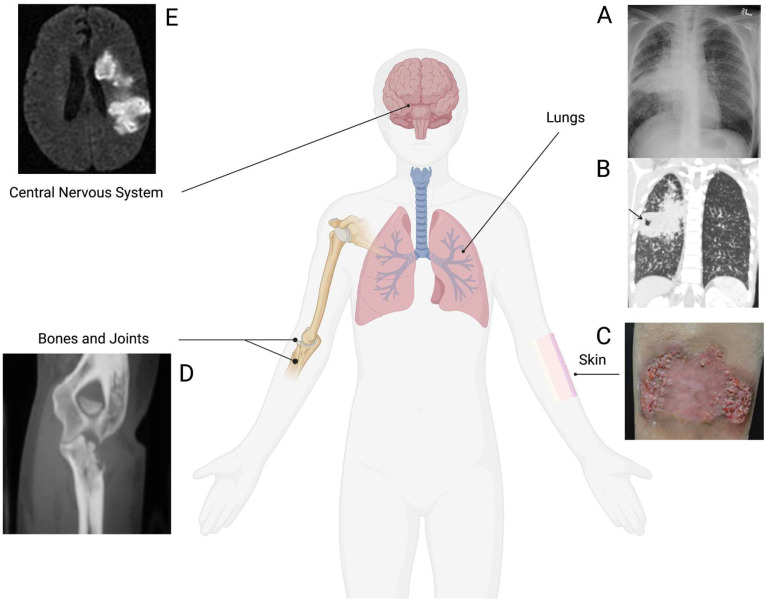
Common sites of *Coccidioides* infection. From the lung, coccidioidomycosis can spread to many different sites, as highlighted above. Chest radiograph and CT image of a patient with pulmonary coccidioidomycosis infection (**A**,**B**), respectively (reproduced from Adam et al. with permission [6]). The arrow in (**B**) points to cavitation where fungal infection is present. Skin infection appearing as verrucous plaques on a Coccidioidomycosis patient (**C**) (reproduced from Garcia et al. with permission [2]). CT image of the patient’s right elbow with eroded ulna due to *Coccidioides* dissemination (**D**) (reproduced from Capoor et al. with permission [11]). MR image showing coccidioidomycosis-induced cerebral infarction (**E**) (reproduced from Lammering et al. with permission [12]). Created using BioRender.com, accessed on 2 February 2023.

**Table 1 jof-09-01002-t001:** Classification of disseminated coccidioidomycosis cases by common sites of dissemination as described in (Adam 2007) [6].

	Site Dissemination Frequency	Frequency of Multisite Infection
Central nervous system	34.3%	19.7%
Blood	21.3%	6.8%
Skeletal-axial	16.9%	20.0%
Skeletal-peripheral	15.5%	18.8%
Soft tissue	7.2%	0.0%
Skin	7.7%	5.9%
Gastrointestinal tract	2.4%	0.0%
Genitourinary tract	1.4%	0.0%
Heart	0.5%	0.0%

**Table 2 jof-09-01002-t002:** Distinctive clinical features of spinal metastasis, vertebral coccidioidomycosis, pyogenic and tuberculous spondylitis.

Variables	Vertebral Coccidioidomycosis	Pyogenic Spondylitis	Tuberculous Spondylitis	Spinal Metastasis
Location on the spine	Thoracic and lumbar	Thoracic and lumbar	Thoracic	Thoracic and lumbar (and rarely cervical)
Associated microorganisms	*Coccidioides* spp.	*Staphylococcus aureus*, Streptococci, Enterococci, *Escherichia coli*, etc.	*Mycobacterium tuberculosis*	No microorganisms associated
Route of spread	Hematogenous	Hematogenous arterial route, direct inoculation from surgery.	Venous, Batson’s paravertebral venous plexus	Venous hematogenous (the most common route), arterial, direct tumor extension, and lymphatic
Vertebral bodies	>1	Few vertebral bodies involved	>1	Few vertebral bodies involved (some rare cases showed > 5 vertebrae involved on the cervical spine
ESP, CRP *	Mildly increased	Markedly increased	Mildly increased	Mildly increased

* ESR, erythrocyte sedimentation rate; CRP, C-reactive protein.

**Table 3 jof-09-01002-t003:** Commonly used antifungal drugs for disseminated *Coccidioides* spp. infection.

Variables	Itraconazole	Fluconazole	Posaconazole	Amphotericin B
Indication	Bone and joint infection Meningitis	Bone and joint infection Meningitis	Severe bone infection	Alternative therapy to azoles Worsening lesions Critical locations such as the vertebral column
Advantage	Good bone penetration	High systemic absorption Low cost Availability in intravenous or oral formulations	Good bone penetration	Good bone penetration
Side effects	Nausea Diarrhea Liver function abnormalities Hypokalemia Hypertension	No general adverse effects Teratogenic for pregnant patients	Generally well tolerated Can cause nausea, vomiting, and hepatotoxicity	Nephrotoxic

## Data Availability

No new data were created or analyzed in this study. Data sharing is not applicable to this article.

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
