# Peer review of "Coccidioidomycosis Osteoarticular Dissemination"

_jof, 2023, doi:10.3390/jof9101002_

Round 1

Reviewer 1 Report (New Reviewer)

In this review, Moni and colleagues attempt to pull together the sparse literature regarding Coccidioides spp dissemination to bone and joints. The article is well organized and addresses an underrepresented focus for disseminated coccidioidomycosis. Sadly, it suffers from two insurmountable issues.

First, much of the manuscript sounds as though it were written by a graduate student trying to do background research for a project they are not familiar with. This feeling is strengthened by the lack of any primary data or images from the authors in the manuscript. Each image of pulmonary or disseminated cocci is reproduced from other publications. The references are old and out of date, ignoring the plethora of publications from the past 5-10 years. Second, most notable in the introduction, is the recitation of statistics gleaned from abstracts of review articles in which the authors present the most extreme value reported. For example, “Skeletal coccidioidomycosis, which accounts for up to 50% of extrapulmonary infections” was derived from the abstract by Janice Blair from 2007 that states “Skeletal coccidioidomycosis occurs in 20% to 50% of disseminated infections.” While using the 50% number sounds much more dramatic, the remainder of the numbers included in this manuscript are small and do not support that 50% (or even 20%) value. Using the data in Table 2, 70/204 cases were skeletal (33%) and if joint involvement is included as well, that number is 40% (84/207). If there are ~350,000 cases per year, with a dissemination rate of 0.5-2%, then even using 0.5%, there would be 1750 cases of DCM/year. If 50% of those are skeletal, that is 875. And yet the case reports and cohort reports cited in this have small numbers (44 patients in Bays, 11 of whom died. That single study was used by the authors to state a 25% mortality rate) and a recent review of all case reports was 92 cases in 87 patients (Koutserimpas, 2022). Introductory facts are incorrect such as “0.5 – 2% of cases progress to a disseminated and life-threatening disease”. Cutaneous skin lesions, especially when they occur below the nares, tend to be isolated and respond well to anti-fungals. They are not life-threatening.

Most of the references are old and out of date – likely copied from the older review articles cited rather than performing a more recent, thorough, review of the literature. For example, the statement that patients of “African descent were mostly associated with increased rates of disseminated coccidioidomycosis but not pulmonary disease” is from a 2001 epidemiology paper. There have been numerous, well controlled, studies out of California and Arizona in the past decade which examine would be more appropriate citations. Likewise, the inclusion of the 1999 paper by Louie et al regarding HLA and ABO blood groups has not been replicated in almost a quarter century. The one-sentence discussion of biologics cites references from 1978, 2004 and 2009. In fact, the median publication date of the references is 2014, only 29 are from the previous 5 years. Performing a PubMed search using (Coccidioides[title/abstract] OR Coccidioidomycosis[Title/abstract] OR "Valley Fever"[Title/abstract]) AND (Skeletal[title/abstract] OR Bone[Title/abstract] OR osteomyelitis[Title/abstract] OR Vertebra*[Title/abstract] OR joint[Title/abstract]) NOT "Rift Valley Fever" produced 240 references including a case series from 2022 by Naeem (41 pediatric musculoskeletal patients), a literature review in 2022 by Koutserimpas (87 cases), a review article on dogs with coccidioides osteomyelitis (Shaver, 2021) – and those were only within the first 18 references. This makes it apparent that the literature "review" was cursory at best.

The discussion of the need for mouse models is clearly written by someone unfamiliar with mouse models of coccidioidomycosis. Without using an attenuated Coccidioides strain, C57/BL6 mice die within 14-21 days after infection from extensive lung damage. That is insufficient time for the mice to develop osteomyelitis. While the DBA strain is more resistant, genetically modified mice, a necessary tool for dissecting immune responses, are usually produced on the B6 background. While the authors state there are no mammalian studies, infections (and publications) exist in dogs, horses, llamas and alpacas to name a few.

For discussion of diagnostics, the authors completely ignored the review from earlier this year (McHardy IH, Barker B, Thompson III GR. Review of Clinical and Laboratory Diagnostics for Coccidioidomycosis, Mycology, 2023, March 8).

Overall, the topic of the paper is valuable, however collaboration with a clinician who cares for patients with Coccidioidomycosis, an understanding of the biology of Coccidioides infections in humans and other mammals, and a thorough, updated review of the literature, is needed to make the manuscript ready for publication. There is no mention of the newer antifungal, Olorofim and there is passing mention of biologics and the risks they pose. What would be very valuable would be a summary of all of the case reports and cohorts. Include a table of all patients in the literature, group however is logical – by site, by duration of infection or treatment, by ethnicity, and then provide an analysis of all of the information.

“Minor” points.

Table 1 is not helpful. The observations listed are not truly clinical features and are too vague to be valuable to a clinician.

Table 2 has misinformation, even from the cited publications. For vertebral bodies under Vertebral Coccidioidomycosis, it states “High number” and yet the range in at least one study was 1-24. There is no reference for the “Always” disc involvement for vertebral Coccidioidomycosis.

The authors state “Cases of coccidioidomycosis with the involvement of both joint and bone (osteoarthritis) are the most severe” – this is incorrect – cases of CNS coccidioidomycosis are the most severe with a high mortality rate. That being said, the mention of relapse in bony disease should be a warning to clinicians not to stop therapy early. Evidence of reactivation of Coccidioides after TNF biologics demonstrates that infections may be clinically absent and immunologically controlled but are not cleared.

Desert rheumatism is another name for Valley Fever, aka Coccidioidomycosis. There is no reason to have that in quotes, nor to have erythema nodosum in quotes.

Author Response

We thank the reviewer for their careful review of our manuscript. 

Reviewer #1

In this review, Moni and colleagues attempt to pull together the sparse literature regarding Coccidioides spp dissemination to bone and joints. The article is well organized and addresses an underrepresented focus for disseminated coccidioidomycosis. Sadly, it suffers from two insurmountable issues.

First, much of the manuscript sounds as though it were written by a graduate student trying to do background research for a project they are not familiar with. This feeling is strengthened by the lack of any primary data or images from the authors in the manuscript. Each image of pulmonary or disseminated cocci is reproduced from other publications.

The purpose of review articles is to summarize the state of the field and it does not require the authors to present their own data. Review articles typically reproduce images from other article with copyright permission.

The references are old and out of date, ignoring the plethora of publications from the past 5-10 years. Second, most notable in the introduction, is the recitation of statistics gleaned from abstracts of review articles in which the authors present the most extreme value reported. For example, “Skeletal coccidioidomycosis, which accounts for up to 50% of extrapulmonary infections” was derived from the abstract by Janice Blair from 2007 that states “Skeletal coccidioidomycosis occurs in 20% to 50% of disseminated infections.” While using the 50% number sounds much more dramatic, the remainder of the numbers included in this manuscript are small and do not support that 50% (or even 20%) value. Using the data in Table 2, 70/204 cases were skeletal (33%) and if joint involvement is included as well, that number is 40% (84/207). If there are ~350,000 cases per year, with a dissemination rate of 0.5-2%, then even using 0.5%, there would be 1750 cases of DCM/year. If 50% of those are skeletal, that is 875.

The statistic has been removed.

And yet the case reports and cohort reports cited in this have small numbers (44 patients in Bays, 11 of whom died. That single study was used by the authors to state a 25% mortality rate) and a recent review of all case reports was 92 cases in 87 patients (Koutserimpas, 2022). Introductory facts are incorrect such as “0.5 – 2% of cases progress to a disseminated and life-threatening disease”. Cutaneous skin lesions, especially when they occur below the nares, tend to be isolated and respond well to anti-fungals. They are not life-threatening.

The term “life-threatening” has been removed.

Most of the references are old and out of date – likely copied from the older review articles cited rather than performing a more recent, thorough, review of the literature. For example, the statement that patients of “African descent were mostly associated with increased rates of disseminated coccidioidomycosis but not pulmonary disease” is from a 2001 epidemiology paper. There have been numerous, well controlled, studies out of California and Arizona in the past decade which examine would be more appropriate citations. Likewise, the inclusion of the 1999 paper by Louie et al regarding HLA and ABO blood groups has not been replicated in almost a quarter century.

We have modified the paragraph about genetic risk factors to include updated references and removed the discussion of HLA and ABO blood groups.

The one-sentence discussion of biologics cites references from 1978, 2004 and 2009.

An expanded discussion of biologics has been included.

In fact, the median publication date of the references is 2014, only 29 are from the previous 5 years. Performing a PubMed search using (Coccidioides[title/abstract] OR Coccidioidomycosis[Title/abstract] OR "Valley Fever"[Title/abstract]) AND (Skeletal[title/abstract] OR Bone[Title/abstract] OR osteomyelitis[Title/abstract] OR Vertebra*[Title/abstract] OR joint[Title/abstract]) NOT "Rift Valley Fever" produced 240 references including a case series from 2022 by Naeem (41 pediatric musculoskeletal patients), a literature review in 2022 by Koutserimpas (87 cases), a review article on dogs with coccidioides osteomyelitis (Shaver, 2021) – and those were only within the first 18 references. This makes it apparent that the literature "review" was cursory at best.

Suggested references have been included. The review study by Koutserimpas was already present in our listed references.

The discussion of the need for mouse models is clearly written by someone unfamiliar with mouse models of coccidioidomycosis. Without using an attenuated Coccidioides strain, C57/BL6 mice die within 14-21 days after infection from extensive lung damage. That is insufficient time for the mice to develop osteomyelitis. While the DBA strain is more resistant, genetically modified mice, a necessary tool for dissecting immune responses, are usually produced on the B6 background. While the authors state there are no mammalian studies, infections (and publications) exist in dogs, horses, llamas and alpacas to name a few.

We have revised the section to state that while reports have been published describing disseminated infection following naturally acquired infection of various domesticated mammalian species, such as dogs, cats, horses, llamas, and alpacas, there is a paucity of experimental small rodent models for studying disseminated infection.

For discussion of diagnostics, the authors completely ignored the review from earlier this year (McHardy IH, Barker B, Thompson III GR. Review of Clinical and Laboratory Diagnostics for Coccidioidomycosis, Mycology, 2023, March 8).

The suggested reference was not available when this review article was originally drafted, we have now included this reference as suggested.

Overall, the topic of the paper is valuable, however collaboration with a clinician who cares for patients with Coccidioidomycosis, an understanding of the biology of Coccidioides infections in humans and other mammals, and a thorough, updated review of the literature, is needed to make the manuscript ready for publication. There is no mention of the newer antifungal, Olorofim and there is passing mention of biologics and the risks they pose. What would be very valuable would be a summary of all of the case reports and cohorts. Include a table of all patients in the literature, group however is logical – by site, by duration of infection or treatment, by ethnicity, and then provide an analysis of all of the information.

A reference for Olorofim has now been added to the section on antifungal treatments.

“Minor” points.

Table 1 is not helpful. The observations listed are not truly clinical features and are too vague to be valuable to a clinician.

Table 1 has been removed.

 Table 2 has misinformation, even from the cited publications. For vertebral bodies under Vertebral Coccidioidomycosis, it states “High number” and yet the range in at least one study was 1-24. There is no reference for the “Always” disc involvement for vertebral Coccidioidomycosis.

I believe the reviewer was referring to table 3, which has been revised.

The authors state “Cases of coccidioidomycosis with the involvement of both joint and bone (osteoarthritis) are the most severe” – this is incorrect – cases of CNS coccidioidomycosis are the most severe with a high mortality rate. That being said, the mention of relapse in bony disease should be a warning to clinicians not to stop therapy early. Evidence of reactivation of Coccidioides after TNF biologics demonstrates that infections may be clinically absent and immunologically controlled but are not cleared.

 We replaced “Cases of coccidioidomycosis with the involvement of both joint and bone (osteoarthritis) are the most severe’ with ‘In addition to CNS coccidioidomycosis, cases of coccidioidomycosis with the involvement of both joint and bone (osteoarthritis) are generally classified as more severe.”

Desert rheumatism is another name for Valley Fever, aka Coccidioidomycosis. There is no reason to have that in quotes, nor to have erythema nodosum in quotes.

Quotations have been removed.

Reviewer 2 Report (New Reviewer)

This appears to be a quite comprehensive review, which should prove useful for the community of Coccidioides scientists and others interested in medical mycology. I like the fact that it focuses on a subject in the coccidioidomycosis realm that is often underrepresented in the review literature and at scientific meetings.

 I have only a few comments for improving the manuscript.

Lines 9-10. While there very well be more than two species of Coccidioides, as far as I know only two have been described. I suggest rewording this passage. Also, as written, “several species of Coccidioides fungi” does not agree with the verb “is”.

Lines 18-21. The sentence beginning “In this review” is very long and difficult to follow. I suggest breaking it into two or more sentences.

Line 30. I suggest “population” instead of “populational.”

Line 61. I suggest “focused on” in place of “were carried on.”

Line 62. Add “The” in front of “main.”

Line 87. Add the word “a” before “few.”

Line 105. The word “species” should not be italicized, and the parenthetical expression “(spp.)” should be deleted. In other places in the article where “spp.” Is used I believe it should not be italicized.

The grammar and style employed in this review are excellent. I noted only a few suggested changes.

Author Response

We thank the reviewer for their suggestions. Please find our response attached.

This appears to be a quite comprehensive review, which should prove useful for the community of Coccidioides scientists and others interested in medical mycology. I like the fact that it focuses on a subject in the coccidioidomycosis realm that is often underrepresented in the review literature and at scientific meetings.

 I have only a few comments for improving the manuscript.

Lines 9-10. While there very well be more than two species of Coccidioides, as far as I know only two have been described. I suggest rewording this passage. Also, as written, “several species of Coccidioides fungi” does not agree with the verb “is”.

These sentences have been revised.

Lines 18-21. The sentence beginning “In this review” is very long and difficult to follow. I suggest breaking it into two or more sentences.

This sentence has been revised.

Line 30. I suggest “population” instead of “populational.”

Revised

Line 61. I suggest “focused on” in place of “were carried on.”

Revised

Line 62. Add “The” in front of “main.”

Revised

Line 87. Add the word “a” before “few.”

Revised

Line 105. The word “species” should not be italicized, and the parenthetical expression “(spp.)” should be deleted. In other places in the article where “spp.” Is used I believe it should not be italicized.

Revised

Reviewer 3 Report (New Reviewer)

Coccidioidomycosis is a recalcitrant disease expanding its endemic territory. No effective vaccines are available. Dissemination to other organs is particularly linked to fatality with this disease which is generally associated with weakened immune system. In this review article MS, the authors focused on particular form of commonly occurring disseminated disease i.e. osteoarticular compiling the existing literature. It is well written and highlights the need of animal model system to study the osteoarticular dissemination. Risk factors, diagnostic, and therapeutic clinical aspects of the pulmonary vs. disseminated disease are also discussed.  Following are the minor comments.

1. Can Table 2 incorporate more than one sites of dissemination among these patients? Whether these patients had concurrent/past pulmonary disease history?

2. Not sure how an animal model system can be developed without other organs complications with this disease.

Author Response

We thank the reviewer for their suggestions on our manuscript. Please find response attached.

Reviewer #3

Coccidioidomycosis is a recalcitrant disease expanding its endemic territory. No effective vaccines are available. Dissemination to other organs is particularly linked to fatality with this disease which is generally associated with weakened immune system. In this review article MS, the authors focused on particular form of commonly occurring disseminated disease i.e. osteoarticular compiling the existing literature. It is well written and highlights the need of animal model system to study the osteoarticular dissemination. Risk factors, diagnostic, and therapeutic clinical aspects of the pulmonary vs. disseminated disease are also discussed.  Following are the minor comments.

  1. Can Table 2 incorporate more than one sites of dissemination among these patients? Whether these patients had concurrent/past pulmonary disease history?

Table 2 has been remade to include multisite frequency. Data on the frequency of concurrent/past pulmonary disease history was not available.

  1. Not sure how an animal model system can be developed without other organs complications with this disease.

The paragraph on animal models has been revised.

Round 2

Reviewer 1 Report (New Reviewer)

The authors have superficially addressed my previous concerns, although there are still basic misunderstandings about coccidioidomycosis. 

This manuscript is a resubmission of an earlier submission. The following is a list of the peer review reports and author responses from that submission.

Round 1

Reviewer 1 Report

This is a broad literature review of coccidioidal skeletal infections.  

Primary suggestions 

In the abstract, the authors state that the basis for their manuscript is their review of case reports of coccidioidal osteomyelitis.  It would be useful to explicitly explain what was their search for case reports and to cite all the published case reports that they found, aggregating those citations all together.  Other parts of the literature review involve case series rather than case reports. Reference 6 is an example.  It would also be useful to list which case series were found in total and their citations. 

 The section on animal studies, starting on line 309, needs considerable improvement. The authors call attention to the fact that there does not exist a literature describing an experimental osteomyelitis model for coccidioidomycosis but do not really provide approaches from other disease models that might prove a useful approach to pursue.  I would suggest that they include what experimental models exist of osteomyelitis for other pathogens and speculate which ones might be applied to the study of coccidioidomycosis.  Without that expansion, there is no reason to include this section at all.

The authors are located at Lawrence Livermore National Laboratories.  Although one, Gabriela Loots, is affiliated with UC Davis Health, her affiliation appears to be a research appointment, and none of the authors are physicians. It is puzzling to me that they are interested in writing a clinical review of this topic.  In the manuscript, management advice is frequently over-simplified to the point of being misleading.  I would suggest that wherever treatment is suggested to be followed, an explicit citation be provided as to the course of that recommendation.

 Additional comments

 In general, refer to the organism as Coccidioides spp, not C. immitis, since in nearly all cases only the genus was determined.  Exceptions might be in the murine studies where the species of the infecting strain is known.

 Line 55: From reference 6: “A preceding illness consistent with primary coccidioidomycosis was noted in 80 patients, 67 of whom were seen at UMC for their initial episode of dissemination. The time from the initial episode of primary coccidioidomycosis was 0-2 months in 41 of the 67 (61%), 3-9 months in 14 (21%), 1-2 years in 3 (4%), and 3 years in 9 (13%) and was as long as 20 years. These data are limited by the fact that only 80 of the 150 patients had an identified syndrome of possible primary coccidioidomycosis, but do suggest that disseminated coccidioidomycosis is frequently preceded by an episode consistent with primary coccidioidomycosis within the preceding 2 months.”  The point here is that most disseminated coccidioidomycosis is established early.  Whether the preceding illness consistent with primary coccidioidomycosis is actually the true point of prior infection is not certain from this report.  From another report (doi: 10.1093/cid/ciaa1154): “In general, dissemination occurred early during the course of disease, with 61% of patients disseminating within the first 2 months of infection in the DCM.”  I would suggest that the manuscript be revised to emphasize that dissemination occurs early and only infrequently occurs years later, usually associated with immunosuppression when it does.

 Line 58:  A better estimate of mortality prior to antifungal therapy can be found in doi: 10.1093/cid/ciaa1154.  Disseminated coccidioidomycosis is not nearly as fatal as portrayed here.  Suggest that these estimates be revised

 Line 217: The role of surgery in patients with coccidioidal skeletal dissemination his highly variable and does not always include aggressive surgery. Suggest this be revised.

 Line 270: Serologic testing is not the gold standard for diagnosing disseminated coccidioidomycosis.  Suggest revising this.

 Line 286-7: High CF titers are often misleading as CF antibody testing is currently done.  See doi: 10.1128/JCM.01318-18 for details and suggest revising this section.

Reviewer 2 Report

Under case reports

Given so many rare reports have been referenced- prosthetic joint infection by coccidioidomycosis should be referenced.

Under risk factors

Recent study by Hsu et al on immunogenetics of coccidioidomycosis dissemination in immunocompetent hosts can be quoted.

Under treatment

1.       Mention the need for therapeutic drug monitoring with itraconazole and posaconazole

2.       Challenges involved in long term azole therapy should be mentioned including toxicities, failure rates and drug interactions

3.       Role of aggressive surgical debridement- especially with vertebral disease in the setting of spine instability, sequestrum, abscess, cord compression needs to be elaborated

4.       Follow up parameters including role of CF titers, symptoms and signs, role of inflammatory markers and imaging (or lack thereof) needs to be mentioned

5.       This is the main study which showed itraconazole as slightly superior to fluconazole for management of skeletal infections and needs to be quoted: “GALGIANI, J.N., A. CATANZARO, G.A. CLOUD, et al. 2000. Mycoses Study Group. Comparison of oral fluconazole and itraconazole for progressive, nonmeningeal coccidioidomycosis: a randomized, double-blind trial. Ann. Intern. Med. 133: 676–686.”; itraconazole and fluconazole remain mainstay of therapy

6.       For has extensive or limb threatening skeletal or vertebral disease, causing imminent cord compromise and for severe osseous disease- initial amphotericin B is recommended followed by oral azoles later- this has not been emphasized; and the need for periodic surgical consultations especially with vertebral disease even if medical therapy alone is initially considered in patients who do not improve; review by LD Herron et al can be quoted.

Is there any role for anti-fungal beads in joint infections with amphotericin or fluconazole?

Line 50, 200: replace coccidioides with Coccidioides.

Line 85 replace cocci with Coccidioides and in any other parts of manuscript.

Line 245: Would remove the word “Viral” from co-infections- viral infection of bone is usually not apparent- the present covid reference is not necessarily viral infection of the bone itself.

Line 268 remove caps for Radiology.

Line 333- sensitivity of PCR from skeletal tissues needs to be discussed with available references if any.

 Line 333- role of Beta-D-glucan in the diagnosis (and caveats) can be addressed. 

Line 340- authors have quoted one study with MRSA co-infection but conclude here that Staph aureus co-infections are very high, enough to mention a need for co-assay; this is not seen widely in clinical practice; if they have literature to support this then this must be referenced in addition to the 1 case report quoted.

 Line 355-357- does not seem to pertain to skeletal coccidioidomycosis for the final concluding statement of an article focusing on the same.